# Maillard Reaction Intermediates and Related Phytochemicals in Black Garlic Determined by EPR and HPLC Analyses

**DOI:** 10.3390/molecules25194578

**Published:** 2020-10-07

**Authors:** Kouichi Nakagawa, Hayato Maeda, Yoshifumi Yamaya, Yudai Tonosaki

**Affiliations:** 1Division of Regional Innovation, Graduate School of Health Sciences, Hirosaki University, 66-1 Hon-Cho, Hirosaki 036-8564, Japan; h17m2222@hirosaki-u.ac.jp; 2Faculty of Agriculture and Life Science, Hirosaki University, Bunkyo-Cho, Hirosaki, Aomori 036-8561, Japan; hayatosp@hirosaki-u.ac.jp; 3Aomori Prefectural Industrial Technology Research Center, Agricultural Products Institute, 91 Kamikita-Gun Rokunohe-Machi Inuotose Yanagisawa, Aomori 033-0071, Japan; yoshifumi_yamaya@aomori-itc.or.jp

**Keywords:** black garlic, Maillard reaction, intermediate, EPR, HPLC, radical

## Abstract

The Maillard reaction intermediates and related phytochemicals in garlic (*Allium sativum* L.), which was heated for various lengths of time, using X-band (9 GHz) electron paramagnetic resonance (EPR) and high performance liquid chromatography (HPLC) were investigated. Non-spin-trap and non-destructive EPR detected the total reaction intermediates (radicals). The g-value of the signal was 2.004. The signal with a peak-to-peak linewidth (ΔHpp) was approximately 0.67 milli Tesla (mT). The values of the intermediates are suggestive of organic compounds. The garlic darkened in color with the increasing number of heating days. Melanoidin, responsible for darkening of the garlic, was detected at an absorbance of 400 nm. Analysis of the correlation between the EPR intensity and melanoidin absorbance showed a good correlation coefficient (0.98). In addition, 5-hydroxymethyl furfural (5-HMF) and total phenolic compounds increased with the increasing number of heating days. Moreover, trace amount of Fe^3+^ was observed in the black garlic by EPR. Non-destructive EPR is a useful method for evaluating not only Maillard reaction intermediates, but also the pigment associated with the reaction processes.

## 1. Introduction

Garlic has been used as a common food throughout the world since ancient times. Approximately three decades ago, black garlic, made from *Allium sativum* L. garlic, came into popularity in Asian countries [1,2]. Black garlic is produced from garlic that is treated with heat and moisture for approximately 14 days. Black garlic contains various nutrients, such as thiosulfates, and total phenols [2,3]. Indeed, previous studies have described the phytochemical-related antioxidant effects and healing benefits of garlic [2,3]. One of the compounds, 5-hydroxymethyl furfural (5-HMF) exhibits antioxidant properties and beneficial effects, such as indicating freshness and safety of honey. However, 5-HMF was shown to exhibit negative effects on human health [4].

The Maillard reaction, which involves the reaction between lipids and sugars with amino acids in garlic and other food upon heating, produces a variety of intermediates and products [5]. The HPLC methodology was proposed recently to evaluate the Maillard reaction intermediates to obtain quality black garlic [6]. Moreover, human taste-active compounds of the Maillard reaction products in roasted garlic were reported [7]. The effect of browning due to sugar and amino acid reaction during the processing of black garlic was also reported [8,9]. Melanoidin and other brown polymer compounds produced from the Maillard reaction give food a characteristic dark brown color. The coloring of black garlic is also attributed to melanoidin, which is a nitrogen-containing heterogeneous polymer compound. If these intermediates are stable, they will remain for a long period of time in the garlic. However, there have been no studies to date examining the relationship between melanoidin or other intermediates and the amount of time garlic is heated when producing black garlic. 

Electron paramagnetic resonance (EPR or electron spin resonance, ESR) is a sensitive and non-destructive technique for measuring paramagnetic species in plants at ambient temperatures [10,11]. EPR/ESR can be used to measure the reaction intermediates and/or paramagnetic species, which contain unpaired electrons such as organic radicals and transition metal ions, in a sample. EPR could provide a useful index for assessing the status of intermediates produced in the Maillard reaction [10]. X-band (9 GHz) EPR measures can detect compounds that contain concentrations of 10^12^ unpaired electron spins per gram [11]. In addition, EPR spectroscopy can provide detailed information about the paramagnetic species, such as phytochemical-related radicals, produced by natural plant antioxidant reactions [12]. 

In particular, regarding antioxidant activity, many studies have investigated the polyphenol content and *S*-allyl-l-cysteine, which is a typical compound in garlic. However, although several research groups [2,3,5] have reviewed the literature on black garlic, there have been no studies examining the Maillard reaction intermediates and products in this sense. These are long-standing unresolved issues. Hence, we sought to analyze the quantitative reaction processes of the Maillard reaction for various food products, including garlic, by EPR. 

In this study, paramagnetic species, colored compounds, and polyphenols in black garlic and garlic were examined using X-band EPR, UV-visible, and HPLC. EPR spectroscopy directly detected paramagnetic species in various garlics. We analyzed 5-HMF and the total phenolic compounds produced during the Maillard reaction, including melanoidin. The total concentration of stable paramagnetic species in several garlic samples, in relation to the amount of time the garlic was heated, was also examined.

## 2. Sample Preparations

### 2.1. Black Garlic Samples

Garlic was harvested in June of 2019 in a farm located at the north (Rokunohe-machi, Aomori Prefecture) of Japan’s Main Island, and was used without additional chemical treatment. After harvesting, the garlic was kept at −2 °C in a freezer until May 2020. The garlic was then maintained in a closed container in a 70 °C incubator for 0, 7, 14, 21, or 28 days. Each of three to four garlic samples were treated for measurements (Appendix A, and Figure 1).

### 2.2. EPR Measurements

A commercially available JEOL RE-3X X-band (9 GHz) EPR spectrometer (JEOL Co. Ltd., Tokyo, Japan) was used for radical measurements. The system was operated in X-band mode at 9.435 GHz using a 100-kHz modulation frequency. EPR values were determined using a JEOL Mn^2+^/MgO standard sample (JEOL Co. Ltd., Tokyo, Japan). All EPR spectra were obtained from a single scan. Typical EPR settings were as follows: microwave power, 5 mW; time constant, 0.1 s; sweep time, 4 min; magnetic field modulation, 0.25 mT; and sweep width, 10 mT. After the measurements, the samples were weighed in order to normalize the EPR intensity for each sample. The number of spins per gram was described elsewhere [10,11]. 

A section of each garlic sample (~2 × mm × 2 mm × 6 mm size, ~0.0235 g) was removed and dried at 5 °C in a refrigerator overnight before EPR measurements. The samples were rolled in pieces of tissue, and inserted into an EPR glass tube (o.d. 5.0 mm, i.d. 4.0 mm, JEOL resonance, Tokyo, Japan) for the sample measurements.

### 2.3. Melanoidin Measurements

Melanoidin content was analyzed with the Hirano et al. method [13]. The skin was removed from the garlic, and the samples were ground with a mill (LAB MILL OSAKA CHEMICAL Co., Ltd., Osaka, Japan). The paste (0.2 g) was extracted with 50 mL water and mixed by vortexing for 10 min. The sample was then centrifuged at 700× *g* for 10 min and the supernatant was filtered through a PVDF (0.45 µm) syringe filter. The sample absorbance at 400 nm was measured using a V-730 BIO UV-visible spectrophotometer (JASCO International Co., Ltd., Tokyo, Japan). 

### 2.4. Total Polyphenol Measurements

The total contents of phenolic compounds for each garlic sample were measured using a modified Folin–Ciocalteu method [14]. Briefly, the skin (peel) was removed from the garlic, and the sample was ground with a mill (LAB MILL OSAKA CHEMICAL Co., Ltd., Osaka, Japan). Then the paste (0.2 g) was extracted with 5 mL of 80% ethanol for 24 h at room temperature. After the supernatant was collected following vortexing and centrifugation at 700× *g* for 10 min, it was filtered through a PVDF (0.45 µm) syringe filter. The 10% Folin–Ciocalteu phenol reagent solution (0.4 mL) (NACALAI TESQUE, INC., Kyoto, Japan) was added to 0.2 mL of the sample solution. After 3 min, 0.8 mL of 10% sodium carbonate was added. The mixture was allowed to stand for 30 min. The absorbance was measured at 750 nm using a V-730 BIO UV-visible spectrophotometer (JASCO International Co., Ltd., Tokyo, Japan). The phenolic compound contents are expressed as pirogallol equivalents.

### 2.5. 5-HMF Quantiication by HPLC and HPLC-ESI-Tof-MS

Each garlic extract was prepared using the same method as for the total polyphenol analysis described above. The HPLC analysis was performed with an ACQUITY UPLC H-Class system (Waters Corporation, MA, USA) using an ACQUITY UPLC HSS T3 column (150 mm × 2.1 mm, 1.8 µm particle size; Waters Corporation, MA, USA). The HPLC mobile phases were 0.1% formic acid [solvent A] and 0.1% formic acid acetonitrile [solvent B]. The linear gradient elution was performed as follows: time t (min), solvent A (%): (0–1 min, 99%), (1–10 min, 99–70%), (10–12 min, 70–35%), and (12–15 min, 99%), with a flow rate of 0.65 mL/min and an injection volume of 1.0 µL. UV−visible (UV/vis) absorption spectra were recorded from 200 to 600 nm using a photodiode-array detector (PDA). The wavelength for UV detection was set at 280 nm. The 5-HMF content was calculated using a standard curve prepared with a 5-HMF (Tokyo Chemical Industry Co., Ltd., Tokyo, Japan). The limit of detection (LOD) was 0.3125 µg, the minimum limit of quantification (LOQ) was 0.625 µg.

The HPLC-ESI(electrospray ionization)-Tof(time-of flight)-MS analysis was performed with a Xevo G2-XS QTof (Waters Corporation, MA, USA). The chromatographic conditions were the same. The HPLC ESI-TOF-MS spectra were acquired by scanning from m/z 100 to 1000 with a capillary voltage of 3.0 kV, cone voltage of 30 eV, and source temperature of 150 °C. Nitrogen was used as a nebulizing gas at a flow rate of 20 L/h. The 5-HMF reagent was purchased from Tokyo Chemical Industry Co., Ltd. (Tokyo, Japan). 

### 2.6. Statistical Analysis

Results are expressed as mean ± standard deviation (SD) or standard error (SE). Statistical analyses between multiple groups were conducted using analysis of variance. Statistical comparisons were made using Tukey‒Kramer tests. Differences were inferred as significant for *p* < 0.05. Analyses were conducted using commercial software (Stat View-J ver. 5.0; SAS Institute Inc., Cary, USA). 

## 3. Results and Discussion

### 3.1. EPR of the Garlic Samples

After heating the garlic, the color of garlic after seven days of heating changed to light brown (Figure 1). The garlic continued to brown with additional days of heating, and the black garlic at 28 days became very soft and juicy. The change in the color of the garlic was likely related to intermediates and/or products of the Maillard reaction.

Figure 2 shows the EPR spectra for garlic and garlic after heating for 14 days, which exhibited a very broad signal at approximately 330 mT. The magnetic field swept from 90 to 390 mT. We did not observe a strong signal across the entire magnetic field for untreated garlic, except for a broad signal around 330 mT. On the other hand, the garlic heated for 14 days showed a strong signal around 336.0 mT, as indicated by the arrow. In addition, there are very weak signals around g ~ 4.11 at low magnetic field for 14 days, which may be related to Fe^3+^. The g-value for Fe-ions can vary depending on the moieties of the compound. Note that oxidation state of iron may have been changed during heating the sample, since there are no detectable iron ions in the untreated garlic spectrum. The concentration of free minerals was reported in a previous study [2]. 

The signal at around 336.0 mT (g ~ 2.00) likely corresponds to intermediate radicals [15,16]. Since unpaired electrons in organic radicals exhibit similar behavior to free electrons, their signals usually appear around g ~ 2.00 region [15,16]. The colored organic intermediates and/or products of the Maillard reactions contain unpaired electrons. Thus, we focused on the signal at around g ~ 2.00 for further analyses of the reaction products. However, origin of the broad signal at approximately 333.0 mT is not known at this point.

Figure 3 shows EPR spectra of a series of garlic samples heated for various lengths of time. The g-value of the signal is 2.004. The signal with a peak-to-peak linewidth (ΔHpp) is approximately 0.67 mT. These values are typical values for stable organic radicals [15,16]. The EPR intensity was analyzed because of the same EPR parameters (g-value and ΔHpp) for all spectra. The number of spins per gram for the signal of the sample heating for 14 days, shown in Figure 3, is approximately 1.28 × 10^17^. The number of heating days and the sample weights are indicated on the left-hand and right-hand sides of the chart, respectively. Four different garlic samples for each heating day were measured. EPR intensity is generally proportional to the amount (weight) of the samples measured. The amount of each garlic sample was not exactly the same when we prepared the garlic samples. Hence, the intensity was normalized, and an average of all measurements was considered. The EPR signal intensities increased with the number of heating days. In addition, the results can be inferred from a relation between dark color of the sample and EPR intensity (Figure 1 and Figure 3).

Figure 4 shows the normalized EPR signal as a function of heating days, where the original EPR intensity was divided by the sample weight to normalize the intensity. The normalized EPR increase at seven days of heating is small, but then rapidly increases up to 28 days. The EPR signal intensity may be associated with the brown color of the garlic, which likely corresponds to melanoidin. Indeed, EPR measures not only the reaction intermediates, but also the products that contain unpaired electrons.

### 3.2. Melanoidin of the Garlic Samples

Figure 5 shows an index of melanoidin in the heated garlic extracts at 400 nm. Generally, melanoidin has a broad absorption from 200 to 500 nm. It has been reported that melanoidin is positively correlated with the absorbance when measured at 400 nm regardless of the molecular weight [13]. The values indicated with different letters were significantly different from one another. The absorbance increased as a function of the number of heating days, and likely corresponds to the EPR signal intensity. Based on the absorbance and EPR measurements, we conclude that melanoidin found in the garlic samples contained unpaired electrons and increased in concentration as a function of the number of heating days.

Figure 6 shows the normalized EPR intensity versus absorbance at 400 nm. There is an excellent correlation coefficient (0.98) between the EPR intensity and the absorbance from seven days to 21 days of heating, and the EPR intensity increase is proportional to the increase in the absorbance. Both values exhibited a 7~10 times increase over the course of the heating experiment. This analysis suggests that melanoidin and other compounds were produced over 21 days. Note that the soft state of the black garlic after 28 days made it difficult to handle, and the EPR intensity was very high at 28 days (Figure 3 and Figure 4). Although Amadori and Heyns contents as the indication of the Maillard reaction were reported [6,8], a positive correlation between EPR intermediate and melanoidin suggests that EPR intensity may be a good index of garlic browning.

### 3.3. 5-HMF of the Garlic Samples

Figure 7 shows the HPLC chromatograms for the various samples at 280 nm. An increase in a specific peak (indicated by an arrow) was observed after 14 days. To identify this peak, it was analyzed by HPLC-ESI-Tof-MS. The maximum absorption was 283.8757 nm in the analysis of the UV/vis absorption spectra (Appendix A, Appendix A). The MS spectrum [M + H]^+^ was 127.0393 (*m*/*z*) and the molecular weight was expected to be 126.11 (Appendix A, Appendix A). Based on these results, the peak was identified as 5-hydroxymethyl furfural (5-HMF), which is a typical intermediate produced in the Maillard reaction. The maximum absorption was 283.8757 nm, and [M + H]^+^ was also in agreement with 127.0397 (*m*/*z*) (Appendix A, Appendix A). Further, the retention time of the HPLC was also matched (Figure 7 and Appendix A). These results show that the peak corresponding to 5-HMF increased with heating time (Scheme 1).

Figure 8 shows the 5-HMF concentration (mg/g) in the various garlic samples. The 5-HMF quantity was calculated using a standard curve prepared using a 5-HMF standard reagent. The increase in 5-HMF was similar to that observed for melanoidin. It has been suggested that 5-HMF is mainly formed from hexose in garlic [13]. 5-HMF has been already reported at 55 °C [4]. However, we reconfirm 5-HMF at the 70 °C heating of garlic. The 5-HMF level increased as the number of heating days increased, except at 28 days. It is very high peak intensity on 28 days heating. In this experiment, 5-HMF, which is an intermediate compound, was typically produced in black garlic. Thus, 5-HMF may be a typical intermediate or end product produced by the Maillard reaction in garlic via this processing method.

### 3.4. Polyphenols in the Garlic Samples

Figure 9 shows total polyphenol contents in the various garlic samples. Polyphenols were found in low concentration at zero and seven days, but it was significantly increased between seven and 14 days. It became about two times on 21 days compared with zero days. The 21 days was almost same as 28 days. Therefore, it was considered that the content of polyphenol reached almost the maximum amount on the 21 days. 

It has been previously reported that black garlic has a higher polyphenol content than raw garlic [2]. The polymerization of polyphenols and decomposition of macromolecules of polyphenols are expected; however, the details remain unclear and further research is needed. The 5-HMF compound is a typical intermediate of Maillard reaction compounds. It has been reported by Lu et al. that black garlic contains 5-HMF [17]. The present study showed that 5-HMF content in black garlic increases with increase in heating time (Figure 8). It has been reported that fructose in black garlic is approximately > 10 times higher than that in raw garlic [2]. It is considered that the increase in monosaccharide levels induces the production of 5-HMF.

With regard to polyphenol and its intermediate, a previous study suggested that the stability of phenoxyl radical intermediate was indicated in the antioxidant sesamol [18]. Thus, we believe that the phenoxyl radical of polyphenol in black garlic may also have contributed to the EPR signal intensity observed in black garlic. 

## 4. Conclusions

The X-band EPR detected reaction intermediates in the various garlic samples, which increased in concentration upon increasing number of days of heating. The present analyses suggested that melanoidin, 5-HMF, and phenolic compounds are produced with various periods of garlic heating. EPR intensity may be a good index of garlic browning. Thus, the EPR and HPLC are very useful techniques that can be applied in the evaluation of Maillard reaction in foodstuffs.

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
