# Peer review of "Maillard Reaction Intermediates and Related Phytochemicals in Black Garlic Determined by EPR and HPLC Analyses"

_molecules, 2020, doi:10.3390/molecules25194578_

Round 1
Reviewer 1 Report
Please see the file attached with comments on the pdf. The issue of absorbance at 400 nm by other compounds than melanoidin h be addressed - the authors have done in the Conclusions, but not in the section dedicated to this aspect.

Author Response
Reviewer 1:
Line 78
- Thank you for your comment. According to your comment, we changed the sentence.
Line 153 Do the authors hypothesise that Fe was present as Fe (0) at the start?
- Thank you for your comment. We do not know all oxidation states in the untreated garlic. We changed the sentence.
Line 191 As in the previous round, oxidized phenolics also absorb in this range. How do the authors support their statement re melanoidin only?
- Thank you for your comment. Reference 13 showed that the 400 nm absorbance has a good correlation with melanoidin of the Maillard reaction. That is the main reason. We changed the sentence
The suggestions were very helpful in improving the original manuscript. We believe that you will find the revision satisfactory. Please let us know your decision concerning suitability for publication.
Thank you.
Sincerely yours,
Kouichi Nakagawa
Kouichi Nakagawa, Professor
Hirosaki University, Japan

Reviewer 2 Report
The manuscript entlited “Maillard Reaction Intermediates and Related Phytochemicals in Black Garlic determined by EPR and HPLC” investigate Maillard’s reaction intermediates and the related polyphenols in black garlic using X-band EPR. Moreover, also polyphenols, melanoidin and 5-HMF were investigated. The manuscript is a resubmitted version, in which several and important changes were made by the authors.
I think that the current version of the paper has definitely improved from the previous one, and I think that now it is suitable for publication in Molecules.
However, I would like to remind the authors of some points:
- Modify the title adding “Maillard Reaction Intermediates and Related Phytochemicals in Black Garlic (Allium sativum) determined by EPR and HPLC analyses”
- Authors need to add the e-mail of each author in affiliation section. Please, check the guidelines of the journal.
- Lane 78, 222, 223, 226, 227 : which supplementary? Please, specify Supporting Figure 1m Supporting Table 1, etc...
Author Response
Reviewer 2:
I think that the current version of the paper has definitely improved from the previous one, and I think that now it is suitable for publication in Molecules.
However, I would like to remind the authors of some points:
- Modify the title adding “Maillard Reaction Intermediates and Related Phytochemicals in Black Garlic (Allium sativum) determined by EPR and HPLC analyses”
-
- Thank you for your suggestion. According to your suggestion, we changed the title.
- Authors need to add the e-mail of each author in affiliation section. Please, check the guidelines of the journal.
- Thank you for your comments. According to your comment, we added all e-mail addresses.
- Lane 78, 222, 223, 226, 227 : which supplementary? Please, specify Supporting Figure 1m Supporting Table 1, etc...
- Thank you very much for your suggest. According to your suggestion, we added the numbers.
The suggestions were very helpful in improving the original manuscript. We believe that you will find the revision satisfactory. Please let us know your decision concerning suitability for publication.
Thank you.
Sincerely yours,
Kouichi Nakagawa
Kouichi Nakagawa, Professor
Hirosaki University, Japan

This manuscript is a resubmission of an earlier submission. The following is a list of the peer review reports and author responses from that submission.
Round 1
Reviewer 1 Report
Please see the document attached.

Reviewer 2 Report
The article by Nakagawa et al. aims at investigating the Maillard reaction intermediates in brown garlic through the application of both EPR and HPLC methodologies. EPR allowed the investigation without extraction process, so could be a useful technique for the monitoring of the process.
However, the manuscript needs major revision before being accepted.
Although in literature there are no works regarding the use of EPR in monitoring the Maillard reaction, some relevant recent references related to the investigation of intermediates and products of Maillard Reaction in brown garlic samples are missing:
- 2-Furoylmethyl amino acids as indicators of Maillard reaction during the elaboration of black garlic. Food Chemistry 2018. https://doi.org/10.1016/j.foodchem.2017.08.016
- Exploring epigallocatechin gallate impregnation to inhibit 5-hydroxymethylfurfural formation and the effect on antioxidant ability of black garlic. LWT - Food Science and Technology 2020 https://doi.org/10.1016/j.lwt.2019.108628
- Effects and mechanism of free amino acids on browning in the processing of black garlic Journal of the Science of Food and Agriculture 2019. https://doi.org/10.1002/jsfa.9707
- Taste-Active Maillard Reaction Products in Roasted Garlic (Allium sativum) . Agric. Food Chem.2016 https://doi.org/10.1021/acs.jafc.6b02396
- An analysis of the changes on intermediate products during the thermal processing of black garlic. Food Chemistry 2018. https://doi.org/10.1016/j.foodchem.2017.06.079
Discussion of the results is missing and should be carried out also mentioning data reported in other works.
Introduction
The scheme reported at page 2 is not described and should be removed.
2.1 Sample preparation
Authors should specify how many garlic were collected (either amount of number). Authors should specify how many garlic they use for 0 days, 7 days, 14 days , 21 days and 28 days time points.
Line 71-74: move to EPR measurements since the authors performed different analyses
2.3 Melanoidin content
How many garlic the authors used to perform this experiment?
2.5 HMF quantification
LOD and LOQ is missing.
Results and Discussion
Figure 3. Sample weights are a mean value? At 21 days sample weight is 0.0087 g? How do the authors explain this value?
Line 177: shows
Line 218-2019: It is very high accumulation on 28 days heating. English not correct.
Line 219: 5-HMF may not be available to react with amino acids. Discuss this period.
Line 219-220: It is noted that color of 5-HMF solution is similar to one for the extracted solution of the black garlic. What do the authors mean? Has the intensity of the color of the two solutions been compared in some way? Or it is just an impression?
Line 220: Thus, 5-HMF may be an intermediate and/or product produced by the Maillard reaction. This finding was already proved in other works (i.e. https://doi.org/10.1016/j.lwt.2019.108628)
3.2.Polyphenols in the garlic samples
Line 227: the first period is not correct. Authors quantified the total polyphenol compounds; they do not perform the identification of polyphenols.
Figure 9: a, b and c are not expressed in the figure.
The authors affirmed in scheme at page 7 the formation of polyphenols and melanoidins as final products. It could be interesting if the authors provide a hypothesis for the formation of these product. How do they originate?
Reviewer 3 Report
The manuscript entitled “Maillard reaction intermediates and related compounds identified in black garlic using EPR and HPLC” investigate Maillard’s reaction intermediates and the related polyphenols in black garlic using X-band EPR. Moreover, also polyphenols, melanoidin and 5-HMF were investigated. The manuscript is generally clear and well written, apart from lack of clarity in some points and minor errors all over the text. It describes a thorough and well-described study, written with some authority. However, before considering it suitable for the publication in Molecules, some small errors need to be corrected.
Introduction: the strength of this article is the combined use of different analytical techniques for the quantification of bioactive compounds contained in black garlic. Indeed, the authors did not limited their research to the simple characterization of the Maillard’s compounds, but also polyphenols, 5-HMF and melanoidin were investigated and quantified. For this reason, I suggest to add additional information in Introduction section concerning the use of black garlic extracts for its phytochemical composition, regarding the polyphenolic profile previously published, uses in traditional and folk medicine, etc…
Materials and Methods:
- Section 2.4: please, report also the original reference of the method.
- Section 2.5. You don’t need specify the acronym in the title of the section. You have to explain it in the text of the section. Please, change the title in “HMF quantification by HPLC-ESI-ToF-MS. Moreover, report a reference for this section. PS, ESI = eLectron spry ionization.
- Moreover, there is no information about the validation of the extraction and analytical method, so it is not known if the extraction method recovers the 100% of the analysed compound. This is one important issue since one of the aim of the manuscript is the quantification of specific compounds from black garlic. So, how can the conclusion be supported without knowing about the recovery of the extraction method?. Therefore, I would recommend the inclusion of a validation of the analytical method, in which, not only the recovery and linearity, but also the matrix effect, limit of detections (LOD), limit of quantification (LOQ), and precision need to be added.
- Has 5-HMF been identified simply basin on retention time?
Results and Discussion: Authors need to compare their data with previously published data.
- Figure 1: please, remove the sentence from the image, and report it in the figure caption. Can authors provide an image with a better quality? For example, with a totally uniform and clear background?
- Figure 5: the quality of this figure is very low. please replace it with higher quality. Moreover, MDPI journals don’t require additional costs for coloured images. For this reason I suggest the authors to re-prepare all the figures. Moreover, the legend reports “Total Melatonin content” meanwhile y-axis reports “Absorbance”. Finally “Values indicated with different letters were significantly different from each other”, but no letters are reported in the graph. Please, introduce them.
- Figure 7: please, remove the grey background from the chromatograms.
References: the article contains only 13 bibliographic references. This is mainly due both to the insufficient introduction, and to the lack of comparison of the data with previously published data.